# The Impact of Stretching Intensities on Neural and Autonomic Responses: Implications for Relaxation

**DOI:** 10.3390/s23156890

**Published:** 2023-08-03

**Authors:** Naoto Imagawa, Yuji Mizuno, Ibuki Nakata, Natsuna Komoto, Hanako Sakebayashi, Hayato Shigetoh, Takayuki Kodama, Junya Miyazaki

**Affiliations:** Department of Physical Therapy, Faculty of Health Science, Kyoto Tachibana University, 34 Yamada-cho, Oyake, Yamashina-ku, Kyoto 607-8175, Japankodama-t@tachibana-u.ac.jp (T.K.);

**Keywords:** stretching, relaxation, point of discomfort, electroencephalography, plethysmography

## Abstract

Stretching is an effective exercise for increasing body flexibility and pain relief. This study investigates the relationship between stretching intensity and relaxation effects, focusing on brainwaves and autonomic nervous system (ANS) activity. We used a crossover design with low- and high-intensity conditions to elucidate the impact of varying stretching intensities on neural activity associated with relaxation in 19 healthy young adults. Participants completed mood questionnaires. Electroencephalography (EEG) and plethysmography measurements were also obtained before, during, and after stretching sessions. The hamstring muscle was targeted for stretching, with intensity conditions based on the Point of Discomfort. Data analysis included wavelet analysis for EEG, plethysmography data, and repeated-measures ANOVA to differentiate mood, ANS activity, and brain activity related to stretching intensity. Results demonstrated no significant differences between ANS and brain activity based on stretching intensity. However, sympathetic nervous activity showed higher activity during the rest phases than in the stretch phases. Regarding brain activity, alpha and beta waves showed higher activity during the rest phases than in the stretch phases. A negative correlation between alpha waves and sympathetic nervous activities was observed in high-intensity conditions. However, a positive correlation between beta waves and parasympathetic nervous activities was found in low-intensity conditions. Our findings suggest that stretching can induce interactions between the ANS and brain activity.

## 1. Introduction

Stretching is an effective exercise for increasing body flexibility and pain relief. Exercise such as yoga and Pilates also involve stretching. Static stretching is a helpful method for enhancing joint range of motion (ROM) [1,2] and has been reported to improve performance [3,4] and reduce injury incidence rates [5,6]. In addition, low-intensity stretching, such as yoga [7], relaxes the autonomic nervous system (ANS). Stretching improves flexibility and relaxation; however, most stretching in physical therapy settings focuses on muscle elongation and ROM improvement.

Essential parameters for maximizing the effects of stretching on body flexibility include intensity, duration, position, and intervention time [8]. The Point of Discomfort (POD) has recently gained attention for determining stretching intensity. POD is the joint angle where muscle stretch or discomfort is experienced without pain. Studies have shown that if POD is not reached, tendon extensibility cannot be achieved [9,10], whereas excessive stretching intensity can increase pain without improving tendon extensibility [10,11]. POD uses the Numerical Rating Scale (NRS), a subjective pain intensity scale, to define stretching intensity. High-intensity stretching corresponds to a POD of 120% or more with NRS 3–6, medium-intensity stretching is a POD of 100–110% with NRS 1–4, and low-intensity stretching is a POD of 100% or less with NRS 0–1. POD is an indicator of setting stretching intensity to enhance tendon flexibility. However, no studies have examined the relationship between POD and relaxation effects.

Objective indicators of relaxation effects include brainwave and ANS activity measurements. Generally, alpha brainwaves increase when relaxed [12], and the parasympathetic nervous system dominates the ANS [13]. Notably, some research has reported that stretching induces parasympathetic nervous system dominance [14]. However, others indicate that sympathetic nervous system activity becomes dominant due to stretching [15], resulting in no consensus. As a result of these conflicting reports, we hypothesized that differences in stretching intensity might affect relaxation effects. Furthermore, the relationship between stretching intensity, brainwaves, and ANS activity has yet to be investigated. In this study, we aimed to reveal the impact of varying stretching intensities on neural activity associated with relaxation effects and the relationship between ANS activity and brain activity during stretching.

## 2. Materials and Methods

### 2.1. Participants

The study included 25 healthy young adults (age 20.7 ± 0.6 years, 14 males, nine females, height 167.9 ± 8.9 cm, and weight 57.8 ± 10.0 kg). Those who experienced orthopedic or neurological symptoms in the lumbar spine, hip, or knee joints or apparent pain in the lower limbs in the previous two weeks were excluded [16]. The study was approved by the Kyoto Tachibana University Ethics Committee (approval number: 23-04), and informed consent was obtained from all participants.

### 2.2. Protocol

Participants completed a questionnaire about mood before stretching (pre). After performing hamstring stretches, they completed the questionnaire again. Brainwave and blood volume pulse (BVP) measurements were obtained before (pre), during, and after (post) stretching to evaluate temporal changes.

This study used a crossover design with two stretching intensity conditions: low-intensity and high-intensity. A 2-day washout period was set between the two stretching sessions [17]. Participants were randomly assigned to start with low-intensity or high-intensity conditions (Figure 1). Low-intensity stretching was defined as a POD < 100% and an NRS of 0–1 [10,11], while high-intensity stretching was described as a POD of 100–110% and an NRS of 3–4 [11].

### 2.3. Two-Dimensional Mood Scale (TDMS)

The TDMS evaluates psychological states based on mood activation and stability [18]. It consists of eight questions with confirmed reliability and validity. A two-dimensional graph with comfort and arousal as axes best illustrates the results. Participants rated their mood on a scale from 0 to 6 for each item, and a formula was used to calculate the scores for stability and activation. The TDMS was administered in this study before and after stretching for each condition to assess mood changes as a relaxation effect.

### 2.4. Visual Analog Scale (VAS)

Mood changes due to stretching were assessed using a 100-mm VAS, with the left end representing a “bad mood” (0 mm) and the right end representing an “excellent mood” (100 mm) [19]. The VAS was measured after each stretching trial, and mood changes due to stretching were assessed over time.

### 2.5. Electroencephalography (EEG)

We used an EEG sensor (Biosignalplux, Plux, Inc., Lisbon, Portugal) with a sampling frequency of 1000 Hz to measure brain activity at Fp1 and Fp2 based on the international 10–20 system. These frontal areas reflect emotional brain functions [20,21,22,23]. A reference electrode was attached to the earlobe, and the sensor was attached after cleaning the electrode attachment sites according to the Biosignalsplux manual.

Participants were instructed to keep their eyes open without focusing on a specific point. Measurements were conducted in a quiet indoor environment and the participants wore short-sleeved tops and bottoms. We controlled the time and room temperature and advised participants to sleep enough and avoid alcohol and caffeine before the measurements. All measurements were conducted between 9 am and 12 pm.

### 2.6. Plethysmography

BVP indicates blood volume in blood vessels and changes with heart rate. By measuring the BVP, we can evaluate changes in heart rate influenced by ANS activity [24]. We used a plethysmography sensor (Biosignalplux, Plux, Inc., Lisbon, Portugal) with a sampling frequency of 1000 Hz, placed on the tip of the right index finger. We used low-frequency (LF) and high-frequency (HF) power values and the LF/HF ratio as indicators of sympathetic and parasympathetic nerve activity, respectively [25].

### 2.7. Joint Angle Measurement

We placed a three-axis accelerometer (TSND-151, ATR-Promotions, Kyoto, Japan) on the femur and tibia to maintain a consistent joint angle during stretching [17]. The accelerometer’s sampling frequency was set to 100 Hz. We used Z-axis angle data to calculate the knee joint extension angle. The difference between the maximum lower leg and the corresponding thigh sensor values was used as the knee joint extension angle.

### 2.8. Stretching

We targeted the hamstring muscle for stretching. The stretching method extended the knee joint while maintaining a 90° hip joint flexion (Figure 2). We tested two stretching intensity conditions (low and high) based on the POD using the NRS. Low-intensity stretching corresponded to NRS 0–1, and high-intensity stretching corresponded to NRS 3–4. Throughout the stretching process, we kept the knee joint extension angle consistent with the respective stretching intensity [26].

Participants were secured to a bed with belts to keep their hip joint flexion stable, and stretching was performed with plethysmography and EEG sensors attached. The stretching protocol included a 60 s resting period before stretching, 120 s stretching, and 30 s rest, repeated five times (Figure 3). Based on previous studies, 120 s was determined to be the minimum necessary hold time for stretching to induce flexibility improvements [27].

### 2.9. Data Analysis

We analyzed EEG and BVP data separately for each phase (stretching and resting). These data were analyzed with a custom-written MATLAB code (v.2023a, MathWorks, Natick, MA, USA). We conducted wavelet analysis on the EEG and BVP data for each stretching and resting phase. This method allowed us to observe temporal fluctuations in different frequency bands. The total measurement time was 810 s.

We filtered brainwaves using a 1–30 Hz bandpass filter and removed blink noise using MATLAB’s automatic artifact removal algorithm. Then, we extracted the alpha (8–12 Hz) and beta (13–25 Hz) bands. We calculated the alpha and beta power values and the alpha/beta ratios. Alpha waves indicate a state of relaxation, whereas beta waves indicate a state of arousal. Dividing the alpha wave by the beta wave provides the alpha/beta ratio, which can be used to indicate a state of relaxation with suppressed cortical activity or a state of attentional focus dominance [20,21,22,23]. These values were log-transformed and used as physiological indicators. We used the average Fp1 and Fp2 for each indicator and calculated each indicator’s left–right ratio (Fp1/Fp2).

We used heart rate variability (HRV) analysis for the BVP. We then extracted the LF (0.05–0.15 Hz) and HF (0.15–0.40 Hz) power values, which were log-transformed as they exhibited exponential variation. We also calculated the LF/HF ratio [28] and the average values of each parameter for each phase.

### 2.10. Statistical Analysis

We checked the normality of the VAS and TDMS questionnaire variables and performed statistical analysis after log-transforming non-normal variables. To examine differences in mood according to stretching intensity, we conducted a repeated-measures two-way ANOVA with intensity (high intensity, low intensity) and trial (VAS)/pre-post (TDMS) factors.

To investigate temporal changes in ANS and brain activity due to stretching intensity, stretch/rest phase, and trial times, we conducted a repeated-measures three-way ANOVA with intensity factor (high intensity, low intensity), phase factor (stretch phase, rest phase), and trial factor (trial times). The repeated-measures three-way ANOVA was performed for ANS activity indicators (LF, HF, and LF/HF ratio) and brainwave activity indicators (alpha power, beta power, alpha/beta ratio, average values for left and right, and left–right ratio). A Bonferroni correction was applied to the ANOVA test results to adjust for the effects of multiple comparisons.

We used Pearson’s correlation analysis to investigate the association between ANS and brainwave activity indicator changes. The statistical analyses were performed with R, ver. 4.2.3. Statistical significance was set at *p* < 0.05.

## 3. Results

### 3.1. Pleasure and Mood Changes during Stretching: VAS and TDMS

No significant effects were found on the trial or intensity in VAS. Stability significantly affected intensity and interaction, increasing in high-intensity conditions (Table 1). No significant effects were observed for activation (Table 1).

### 3.2. ANS Activity Changes during Stretching

Significant effects were found in LF, with higher values during the rest phase than in the stretch phase. No significant effects were observed on HF. The LF/HF ratio revealed significant effects on phase and trial, with higher values during rest (Table 2, Figure 4).

### 3.3. Brainwave Activity Changes during Stretching

Significant effects were observed for each phase and trial on alpha power, with higher values during the rest phase than in the stretch phase. Interaction effects were also found. Beta power showed significant effects in these phases, with higher values during rest. No significant effects were found on the alpha/beta ratios (Table 2, Figure 5).

In left–right ratios, significant effects were not observed on alpha and beta power. In the alpha/beta ratio, the stretch/rest phases were significantly affected, with higher values during the stretch phase (Table 2, Figure 5).

### 3.4. Correlation between ANS Activity and Brainwave Activity

In high-intensity conditions, we found significant negative correlations between the resting phase LF/HF ratio, stretch phase alpha activity, and rest phase alpha activity (Table 3). We also observed a significant negative correlation between rest phase LF and stretch phase alpha activity (Table 3).

A significant positive correlation was found between stretch phase LF and stretch phase beta activity in low-intensity conditions. By contrast, a significant positive correlation was observed between stretch phase HF and rest phase beta activity (Table 3). 

## 4. Discussion

In this study, we explored differences in temporal changes in ANS and brain activities during stretching of varying intensity levels. We also investigated the relationship between these changes. We found no significant differences in ANS activity based on stretching intensity. However, sympathetic nervous activity (represented by the LF/HF ratio) showed higher activity during the resting phase than in the stretching phase. Regarding brain activity, alpha and beta waves showed higher activity during the resting phase than in the stretching phase. We observed a negative correlation between alpha wave activity and the LF/HF ratio under high-intensity conditions. By contrast, a positive correlation between beta wave activity and HF was found under low-intensity conditions.

When focusing on temporal changes in ANS activity caused by stretching, we observed higher LF and LF/HF ratio values during the rest phase than in the stretch phase. LF represents both sympathetic and parasympathetic nervous activity, the LF/HF ratio represents sympathetic activity, and HF represents parasympathetic activity [25]. Consequently, our findings suggest that increased sympathetic nervous activity after stretching occurs regardless of stretching intensity differences. Research has reported that static stretching increases heart rate and blood pressure [29,30]. Underlying mechanisms have been linked to muscle sympathetic nerve activity, cardiac sympathetic nerve activity, and vagal withdrawal [31,32,33]. In our study, sympathetic nervous activity increased during the resting phases after stretching, which deviates from some previous findings. Our wavelet analysis, which offers better time resolution than the Fast Fourier Transform analysis used in previous studies, may have contributed to these differences. This study’s wavelet analysis allowed us to identify characteristic differences between the stretch and rest phases. The increased LF/HF ratio during the rest phases after stretching suggests that increased muscle sympathetic nerve activity caused by stretching may persist after rest. However, other research has reported that stretching leads to the dominance of parasympathetic nervous activity [7,34], particularly during full-body stretches such as yoga. By contrast, we focused on localized hamstring stretching, which may have affected our results. Moreover, the stretching time in this study was longer than in previous research, and the difference in stimulation time to the body may have influenced the outcomes.

This study examined temporal changes in brainwave activity during stretching, focusing on alpha and beta waves in the frontal region. No differences in brain activity were observed based on stretching intensity. However, the rest phase increased compared to the stretch phase. Alpha waves represent a “standby state” in brain activity [35] and are also indicators of self-reflection [36]. The increase in alpha wave activity during the rest phase suggests a heightened self-reflective state when external stimuli are absent. Furthermore, the left–right dominance analysis based on frequency band ratios showed higher alpha/beta ratios in the Fp1 region during the stretch phase than in the rest phase. The Fp1 and Fp2 regions are involved in the frontal area’s emotional processing, with the left hemisphere associated with positive emotions and the right hemisphere with negative emotions [37]. Our results suggest that stretching increases brain activity related to positive emotions, leading to a relaxed state after stretching. Beta waves are associated with directing attention or emotional processes [38]. Therefore, observed changes in brainwave activity due to stretching indicated a state of increased self-reflection and evoked positive emotions during the rest phase.

Our study is the first to explore the relationship between brainwave and ANS activities during stretching, focusing on stretching intensity. High-intensity stretching revealed a significant negative correlation between alpha wave activity and sympathetic nervous activity (LF/HF ratio) during the stretch and rest phases. Consistent with prior research, increased alpha wave activity was associated with suppressed sympathetic nervous activity [36]. By contrast, low-intensity stretching showed significant positive correlations between beta wave activity, sympathetic nervous activity (LF), and parasympathetic nervous activity (HF). Similar to previous findings [39], we found a significant positive correlation between beta wave activity and HF. Beta wave activity has been reported to be related to non-invasive somatosensory stimuli intensity, such as touch [40]. In addition, beta wave activity in the frontal area increases during pleasant touch stimuli compared to unpleasant touch stimuli [41]. Differences in the correlation between beta wave activity and LF and HF depending on stretching intensity could be due to the relationship between stimulus intensity and beta wave activity. Therefore, our study results suggest that brain activity is associated with sympathetic nervous activity during high-intensity stretching, and parasympathetic nervous activity during low-intensity stretching. These findings imply that brainwave and ANS activity interaction may vary depending on stretching intensity.

Our study is the first to analyze how differences in stretching intensity affect temporal changes and relationships between brainwave and ANS activities. When determining signs of stretching, alpha and beta wave activities and sympathetic nervous activity (LF/HF ratio) were higher during rest than during stretching. Increased left hemisphere alpha wave activity suggested that positive emotions are associated with brain activity during stretching [37]. When focusing on emotional changes induced by stretching, high-intensity stretching-induced stability (TDMS) increased, indicating relaxation effects. High-intensity stretching revealed a correlation between alpha wave activity and sympathetic nervous activity (LF/HF ratio). These findings suggest that stretching has common neurophysiological effects, with additional effects depending on the stretch intensity.

The impact of stretching on brainwave and ANS activity in this study can be explained from a neurophysiological perspective. The neurovisceral integration model, which explains the central nervous system’s control of ANS activity, proposes that the pre-frontal cortex forms the prefrontal–amygdala pathway [42]. This pathway’s inhibition from the prefrontal cortex to the amygdala is crucial for heart rate variability control [43]. Furthermore, the medial prefrontal cortex forms a cortico-subcortical autonomic network with sympathetic motor neurons through the hypothalamus and locus coeruleus. This network also forms a pathway with subcortical regions such as the caudate nucleus. This network transmits information to the medulla about internal states such as anxiety and arousal and pleasant or unpleasant stimuli, which is then reflected in ANS activity [44]. Our study results suggested that stretching stimulation and differences in stretching intensity altered arousal and pleasant emotions in the prefrontal cortex. In addition, these stretching effects may affect ANS activity through the prefrontal–amygdala pathway and the cortico–subcortical autonomic network.

This study has a few limitations. Firstly, brain wave measurements were only obtained from the prefrontal regions Fp1 and Fp2. Similar research shows brain wave activity changes in the frontal and parietal regions in response to stimuli [45,46]. Therefore, this study’s stretching activities may also be linked to activity changes in these regions. Secondly, our study only included healthy young adults, so brain waves and autonomic nervous system activity may differ based on age or pain levels. Thirdly, we did not assess participants’ flexibility, which might have influenced the results due to varying physical stress levels. Future research should explore different participant groups to better understand the application of stretching in clinical settings and its neurophysiological effects. Examining a broader range of participants and brain wave measurement locations will provide more comprehensive insights into stretching interventions’ potential benefits and mechanisms.

## 5. Conclusions

Our study results suggest that stretching can change ANS and brain activity regardless of stretching intensity. Additionally, there appears to be a connection between brain and ANS activity while applying stretching stimuli. This connection implies that the brain may regulate ANS activity at rest and when exposed to external stimuli.

## Figures and Tables

**Figure 1 sensors-23-06890-f001:**
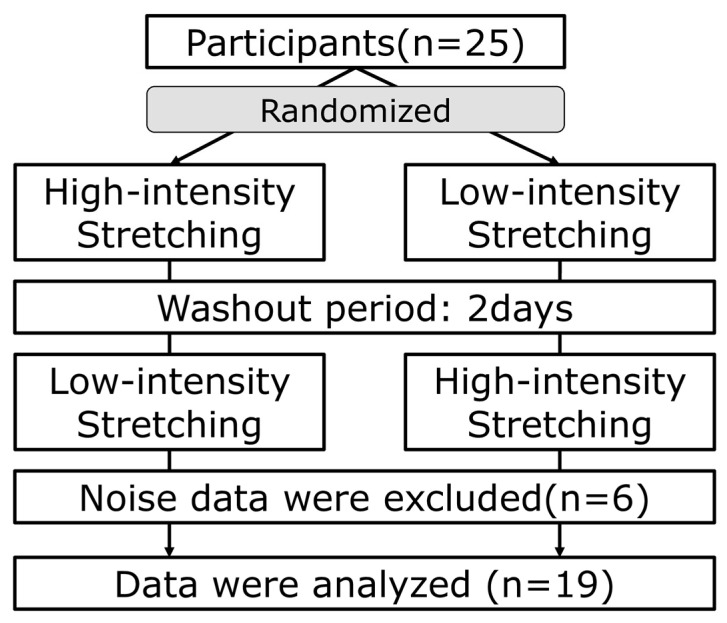
Study protocol. Twenty-five participants were randomly divided into two conditions: Those who started with high-intensity stretching followed by a 2-day washout period, and those who started with low-intensity stretching followed by a 2-day washout period, followed by high-intensity stretching. Nineteen participants were analyzed out of twenty-five participants, excluding those with inaccurate measurements of ANS and brainwave activity due to noise.

**Figure 2 sensors-23-06890-f002:**
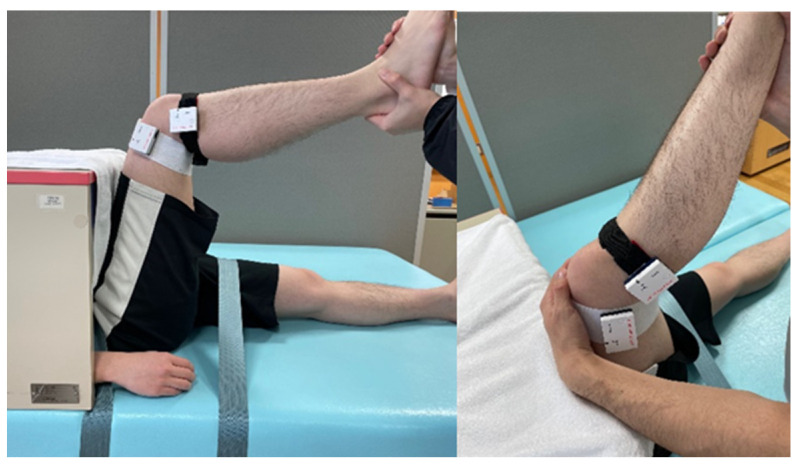
Stretching position. The examiner grasped the participant’s lower leg and stretched it to extend the knee joint. Accelerometers were affixed to the distal thigh and proximal lower leg.

**Figure 3 sensors-23-06890-f003:**
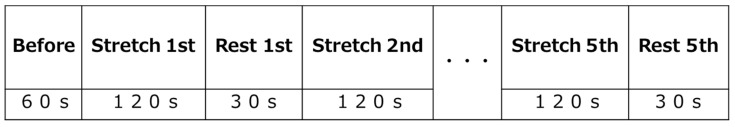
Stretching protocol.

**Figure 4 sensors-23-06890-f004:**
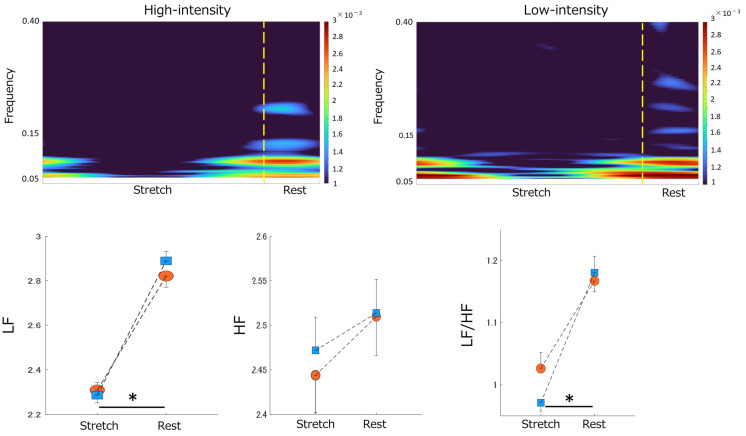
Stretching−elicited time-frequency responses to ANS activity by differences in stretching intensity. Top panel: The color scale represents the power magnitude relative to the difference in phase factor (Stretch/Rest). The left panel indicates the time-frequency response to High−intensity stretching. The right panel displays the time−frequency response to Low−intensity stretching. Bottom panel: The effect of differences in stretching intensity on the magnitude of the LF power (bottom left), HF power (bottom center), and LF/HF ratio (bottom right) by differences in phase factor. Blue squares indicate Low−intensity stretching conditions. Orange circles indicate High−intensity stretching conditions. “*” indicates a significant difference between the stretch and rest phases (*p* < 0.05). Data are mean ± SEM. ANS, Autonomic Nervous System; LF, Low Frequency; HF, High Frequency.

**Figure 5 sensors-23-06890-f005:**
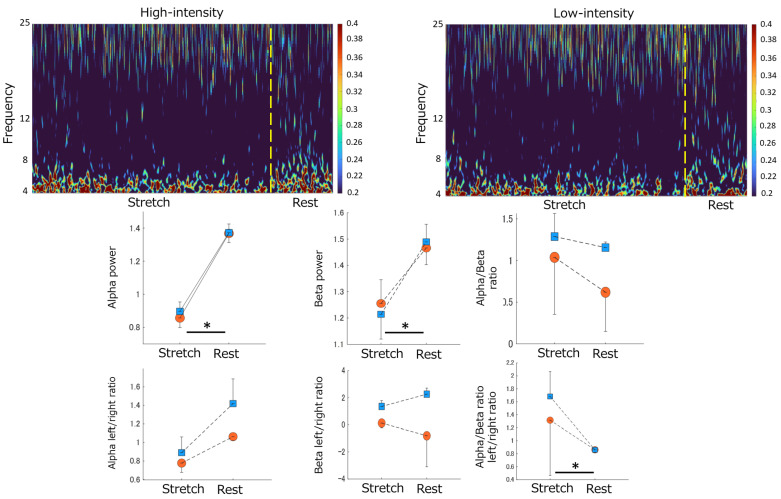
Stretching-elicited time-frequency responses on EEG by differences in stretching intensity. Top panel: The color scale represents the power magnitude relative to differences in phase factor (Stretch/Rest). The left panel indicates the time-frequency response of High-intensity stretching. The right panel displays the time-frequency response of Low-intensity stretching. Middle panel: The effect of differences in stretching intensity on the magnitude of the Alpha power, Beta power, and Alpha/Beta ratio by differences in phase factor (from left to right). Bottom panel: The effect of differences in stretching intensity on the magnitude of the Alpha left/right ratio, Beta left/right ratio, and Alpha/Beta ratio left/right ratio by differences in phase factor (from left to right). Blue squares indicate Low-intensity stretching conditions. Orange circles indicate High-intensity stretching conditions. “*” indicates a significant difference between the stretch and rest phases (*p* < 0.05). Data are mean ± SEM. EEG, Electroencephalography.

**Table 1 sensors-23-06890-t001:** Repeated-measures two-way ANOVA for VAS and TDMS results on stretching.

	Intensity	Trial (VAS)/ Pre-Post (TDMS)	Intensity × Trial (VAS)/ Pre-Post (TDMS)
VAS	*p* = 0.82	*p* = 0.87	*p* = 0.23
	η^2^ ≤ 0.01	η^2^ ≤ 0.01	η^2^ ≤ 0.01
TDMS			
Stability	*p* = 0.001 **	*p* = 0.41	*p* = 0.001 **
	η^2^ = 0.18	η^2^ ≤ 0.01	η^2^ = 0.25
Activation	*p* = 0.47	*p* = 0.62	*p* = 0.75
	η^2^ ≤ 0.01	η ^2^ ≤ 0.01	η^2^ ≤ 0.01

VAS, Visual Analog Scale; TDMS, Two-Dimensional Mood Scale. **: *p* < 0.01.

**Table 2 sensors-23-06890-t002:** Repeated-measures three-way ANOVA for HRV and EEG results on stretching.

Variable	Intensity	Trial	Phase	Intensity ×Trial	Intensity ×Phase	Trial ×Phase	Intensity ×Trial ×Phase
HRV							
LF	*p* = 0.76	*p* = 0.69	*p* = 0.001 **	*p* = 0.40	*p* = 0.14	*p* = 0.92	*p* = 0.27
	η^2^ ≤ 0.01	η^2^ ≤ 0.01	η^2^ = 0.33	η^2^ ≤ 0.01	η^2^ ≤ 0.01	η^2^ = 0.011	η^2^ ≤ 0.01
HF	*p* = 0.83	*p* = 0.74	*p* = 0.22	*p* = 0.23	*p* = 0.50	*p* = 0.65	*p* = 0.33
	η^2^ ≤ 0.01	η^2^ ≤ 0.01	η^2^ ≤ 0.01	η^2^ = 0.010	η^2^ ≤ 0.01	η^2^ ≤ 0.01	η^2^ ≤ 0.01
LF/HF ratio	*p* = 0.39	*p* = 0.64	*p* = 0.001 **	*p* = 0.29	*p* = 0.071	*p* = 0.76	*p* = 0.035 *
	η^2^ ≤ 0.01	η^2^ ≤ 0.01	η^2^ = 0.15	η^2^ ≤ 0.01	η^2^ ≤ 0.01	η^2^ ≤ 0.01	η^2^ = 0.016
EEG							
Fp1 + Fp2 average							
α power	*p* = 0.72	*p* = 0.001 **	*p* = 0.001 **	*p* = 0.81	*p* = 0.55	*p* = 0.047 *	*p* = 0.46
	η^2^ ≤ 0.01	η^2^ = 0.050	η^2^ = 0.18	η^2^ ≤ 0.01	η^2^ ≤ 0.01	η^2^ ≤ 0.01	η^2^ ≤ 0.01
β power	*p* = 0.92	*p* = 0.19	*p* = 0.0082 **	*p* = 0.73	*p* = 0.40	*p* = 0.36	*p* = 0.35
	η^2^ ≤ 0.01	η^2^ ≤ 0.01	η^2^ = 0.024	η^2^ ≤ 0.01	η^2^ ≤ 0.01	η^2^ ≤ 0.01	η^2^ ≤ 0.01
α/β ratio	*p* = 0.26	*p* = 0.43	*p* = 0.46	*p* = 0.53	*p* = 0.70	*p* = 0.22	*p* = 0.84
	η^2^ ≤ 0.01	η^2^ = 0.015	η^2^ ≤ 0.01	η^2^ = 0.012	η^2^ ≤ 0.01	η^2^ = 0.011	η^2^ ≤ 0.01
Left/Right ratio							
α power	*p* = 0.06	*p* = 0.27	*p* = 0.083	*p* = 0.28	*p* = 0.66	*p* = 1.0	*p* = 0.62
	η^2^ ≤ 0.01	η^2^ ≤ 0.01	η^2^ = 0.016	η^2^ = 0.012	η^2^ ≤ 0.01	η^2^ ≤ 0.01	η^2^ ≤ 0.01
β power	*p* = 0.16	*p* = 0.51	*p* = 1.0	*p* = 0.31	*p* = 0.38	*p* = 0.27	*p* = 0.65
	η^2^ ≤ 0.01	η^2^ ≤ 0.01	η^2^ ≤ 0.01	η^2^ = 0.012	η^2^ ≤ 0.01	η^2^ = 0.015	η^2^ ≤ 0.01
α/β ratio	*p* = 0.46	*p* = 0.50	*p* = 0.037 *	*p* = 0.76	*p* = 0.36	*p* = 0.46	*p* = 0.79
	η^2^ ≤ 0.01	η^2^ = 0.011	η^2^ ≤ 0.01	η^2^ ≤ 0.01	η^2^ ≤ 0.01	η^2^ = 0.011	η^2^ ≤ 0.01

HRV, Heart Rate Variability; EEG, Electroencephalography; LF, Low Frequency; HF, High Frequency. **: *p* < 0.01. *: *p* < 0.05.

**Table 3 sensors-23-06890-t003:** Correlation analysis between ANS and EEG activity for the stretch and rest phases.

		LF Stretch	LF Rest	HF Stretch	HF Rest	LF/HF Ratio Stretch	LF/HF Ratio Rest
Low− intensity	Fp1 + Fp2 average						
	α power Stretch	0.14	−0.14	0.20	0.15	−0.09	−0.41
	α power rest	0.15	−0.22	0.20	0.1	−0.09	−0.33
	β power Stretch	0.47 *	0.22	0.42	0.34	−0.10	−0.07
	β power rest	0.41	0.24	0.50 *	0.37	−0.3	0.01
	α/β ratio Stretch	0.01	0.07	0.10	0.13	−0.12	−0.09
	α/β ratio rest	−0.23	−0.33	−0.23	−0.25	0.17	−0.33
	Left/Right ratio						
	α power Stretch	0.14	0.26	0.03	0.05	0.11	0.19
	α power rest	0.08	0.29	0.01	0.03	0.02	0.19
	β power Stretch	0.16	−0.19	−0.02	−0.06	0.13	0.06
	β power rest	0	0.10	0.03	−0.06	−0.21	0.04
	α/β ratio Stretch	−0.04	−0.15	−0.04	−0.06	−0.07	−0.18
	α/β ratio rest	0.1	−0.04	0.11	0.09	0.12	−0.09
High− intensity	Fp1 + Fp2 average						
	α power Stretch	−0.26	−0.51 *	0.12	−0.14	−0.28	−0.61 **
	α power rest	−0.08	−0.40	0.30	0	−0.35	−0.65 **
	β power Stretch	−0.17	−0.28	0.26	0.03	−0.18	−0.42
	β power rest	−0.03	−0.19	0.33	0.09	−0.18	−0.37
	α/β ratio Stretch	0.15	0.14	0.07	0.11	−0.04	−0.01
	α/β ratio rest	−0.33	−0.4	−0.05	−0.23	−0.12	−0.20
	Left/Right ratio						
	α power Stretch	−0.18	−0.10	0.15	0.04	−0.19	−0.22
	α power rest	0.33	0.32	0.12	0.28	0.01	−0.01
	β power Stretch	−0.25	−0.19	0.03	−0.11	−0.08	−0.09
	β power rest	−0.31	−0.36	−0.03	−0.22	−0.11	−0.16
	α/β ratio Stretch	0.11	0.19	0.05	0.14	0	0.03
	α/β ratio rest	−0.29	−0.26	−0.11	−0.12	0.09	−0.16

ANS, Autonomic Nervous System; EEG, Electroencephalography; LF, Low Frequency; HF, High Frequency. **: *p* < 0.01. *: *p* < 0.05.

## Data Availability

The data presented in this study are available on request from the corresponding author. The data are not publicly available due to privacy and ethical restrictions.

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
