# Peer review of "The Impact of Stretching Intensities on Neural and Autonomic Responses: Implications for Relaxation"

_sensors, 2023, doi:10.3390/s23156890_

Round 1

Reviewer 1 Report

This is an interesting paper that measured EEG and HRV differences between high and low intensity stretching. I have a few concerns that need to be addressed.

1) IFCN guidelines recommend filtering 1 to 50 or 60 Hz-why have you chose 1 to 30 Hz?

2) Why did you log transform the alpha and beta power values?

3) Why did you log transform the LF and HF HRV data?

I understand transforming data for statistical comparison, but when you want to understand physiology, how do such transformations even make sense?

4) The authors do not indicate how p values were adjusted to account for multiple statistical comparisons

I like the fact that the authors used point of discomfort to test two stretching protocols (high vs low intensity), but much better justification of why there durations were chose and also why passive vs active stretching are needed in introduction and discussion.

Reviewer 2 Report

This manuscript addresses the relationship between stretching intensity and relaxation effects. The authors demonstrate stretching can induce interaction between the ANS and brain activity. This is an interesting study that focused on the stretching intensity affecting the ANS and brain activity. In my opinion it requires major revision before ready for publication. I have found a few issues that, once addressed, will improve the manuscript.

Major points

1. The purpose of this study was to reveal the impact of various stretching intensities on neural activity associated with relaxation effects. However, the authors concluded that the relationship stretching can change ANS and brain activity. The purpose and conclusion are not consistent, and the Introduction should be rewritten.

2. I don't understand why the authors focused on the relationship between POD and relaxation effect. Isn't POD an indicator for improving tendon flexibility?

3. Alpha waves are suppressed when eyes are open. How did the authors identify the alpha waves during the EEG measurement with the eyes open? Even if the eyes were open, there would be blink noise in Fp1 and Fp2. How did you remove this artifact? 

4. Why didn't the authors do phase statistics on TDMS? 

5. There is no description of statistical methods for correlating ANS activity with brainwave activity. 

6. Figure 4: The blue square and orange circle seem to be significant in the stretch phase. Do they influence the difference between stretch and rest? 

7. Table3: Why is there a difference in the number of items for the low intensity and the high intensity? 

8. The purpose of this study was to investigate how the EEG and ANS change with different stretching intensities. What is the purpose of using TDMS? 

9. The authors stated ‘Finally, some reports suggest that LF may also represent parasympathetic nervous activity [34], indicating that the effects of stretching on ANS activity in this study could be related to increased parasympathetic activity.’

Since the change in LF/HF was significant, it would seem that the change in LF mostly reflects sympathetic activity. Even if the HF did not change, this conclusion is too far-fetched.

10. Discussion is too long and difficult to understand the point of the discussion.

Stretching Intensity, Autonomic Activity and EEG Changes

Relationship between autonomic activity and EEG changes

Relationship between relaxation and stretch intensity

The discussion should be rewritten to focus on the above points.

Minor points

1. There are 25 examples of attendees in the method, but there are 23 attendees in the figure 1.

2. The figure 1 should present the final number of cases analyzed.

3. Add explanation to the figure 1.

4. Specify the name, version, and vendor of the software that generated the statistics.

5. Figures: What do the blue squares and orange circles indicate? What is the * mark?

6. Line 264: ‘however, these waves increased during the rest phase after the stretch phase compared to the stretch phase.’ You mean “However, there was an increase in these waves during the rest phase in comparison to the stretch phase.”

Round 2

Reviewer 2 Report

This second version of the paper is improved, the authors are to be commended.